# OpenReview forum: "Enhancing Neural Network Transparency through Representation Analysis"
_ICLR.cc/2024/Conference — Submitted to ICLR 2024_

### Official Review · Reviewer_ZgPm · 2023-10-29

**Soundness:** 2 fair
**Presentation:** 2 fair
**Contribution:** 3 good
**Rating:** 5
**Confidence:** 3

**Summary:**

The authors propose to understand a network's computation and dynamics for honesty through representational analysis. Once the representation is characterized, the authors propose three tools to control or augment the model's output to improve its honesty.

**Strengths:**

- The motivation of understanding model's computation through representational analysis, similar to neuroscience, might benefit the deep learning community.
- The approach to improve model performance using a lower dimension representation and developing suitable control techniques is novel.

**Weaknesses:**

- Unclear implementation of RepE: While the authors have described the LAT and control baselines, it helps the reader to understand how these methods are implemented if the mathematical notations are included. Specifically, what model manipulations contribute to the performance increase in Table 1 & 2, and Figure 3? How do you exactly analyze the representations? What do the values as performance mean? The paper is written well as a descriptive or narration but omits details about how a pretrained model can be controlled "The tuned low-rank matrices can serve as controllers, which can be merged into the model weights."


- Inappropriate manuscript style for ICLR: The conference manuscript is written in a style more suited for a journal publication rather than for ICLR. Most of the implementation details, results and network analysis are missing in the 9 pages and are left to the supplementary material. While the motivation of the manuscript is relevant, the manuscript is not informative enough as an ICLR proceeding.

- Minor: Figure 1 is unclear to describe the difference between the bottom-up (mechanistic) vs top-down (normative) approaches of explaining model computation. The classification of top-down as normative vs bottom up as mechanistic from computational neuroscience [1] seems to be in contradiction with the terminology used in the paper?


[1] Levenstein, D., Alvarez, V. A., Amarasingham, A., Azab, H., Chen, Z. S., Gerkin, R. C., ... & Redish, A. D. (2023). On the role of theory and modeling in neuroscience. Journal of Neuroscience, 43(7), 1074-1088.

**Questions:**

- will the code to analyze pretrained models be publicly available?

---

> ### Author Response · Authors · 2023-11-21
> **Author Response**
>
> Thank you for your careful analysis of our work. We hope the following response addresses your concerns.
>
> **Clarified method section in main paper.**
>
> Due to the page limit, many of the details of our methods are in Appendix B. For example, we include a full specification of our LoRRA method in Algorithm 1. However, we agree that readers would benefit from more details in the main paper. We have updated Section 3 in the main paper to give a more detailed and technical description of our reading and control methods. Some details are still left to the appendix, but the main paper is more self-contained now. Thank you for your suggestion.
>
> In Appendix D.1, we include several ablations of our reading and control methods. We also compare the effect of reading vectors vs. contrast vectors vs. LoRRA in Table 2. For all tables, the meaning of the values is indicated in the text. Specifically, Tables 1 and 2 display TruthfulQA MC1 accuracy values as percentages.
>
> **Expanded summary of the appendix in main paper.**
>
> > Most of the implementation details, results and network analysis are missing in the 9 pages and are left to the supplementary material.
>
> We agree that this reduced readability in the original submission. Thanks to your suggestion, we have made significant improvements to the readability of the main paper, which is now much more self-contained. For example, we rewrote parts of Section 3 to give a more detailed and technical description of our method’s implementation in the main paper. We also expanded the summary of the appendix in the main paper (Section 5). This summary now gives the major takeaways from each section of the appendix. With these improvements, we hope your concerns regarding the paper format have been addressed. If we have addressed the thrust of your concerns, we kindly ask that you consider raising your score.
>
> **Neuroscience terminology.**
>
> Thank you for pointing us to the neuroscience paper from Levenstein et al. (2023). Their terminology is fully compatible with our usage; We both use “mechanism” to refer to bottom-up, low-level understanding of a system. Our notion of the top-down “representational” approach to studying systems does not have an analogue in their work, although their use of “descriptive” analysis is somewhat close to what we mean. Rather, we are basing the dichotomy in Figure 1 on the notion of levels of analysis in complex systems theory discussed in Gell-Mann (1995), and on the Hopfieldian and Sherringtonian distinction in cognitive neuroscience, which was identified by Barack and Krakauer (2021). Thus, their terminology is not in contradiction with our own.

---

> > ### Comment · Reviewer_ZgPm · 2023-11-21
> >
> > I thank the authors for making the manuscript more self-contained. I have revised the score to 5 but no more.

---

### Official Review · Reviewer_gKW5 · 2023-11-05

**Soundness:** 3 good
**Presentation:** 2 fair
**Contribution:** 3 good
**Rating:** 6
**Confidence:** 2

**Summary:**

This paper introduces an approach called representation engineering (RepE) which enables a better understanding of large language models. The authors showcase a wide range of applications of RepE, with a focus on honesty.

**Strengths:**

- As language models get larger, devising techniques that enable a better understanding of its dynamics are important for transparency and control. While most recent approaches have focused on mechanistic interpretability, this work breaks down the representation engineering approach which is top-down. The two approaches are contrasted in a clear way in the main text.
- A very extensive set of experiments are conducted, and the results support the authors' claims about the insightfulness of the proposed approach. The method provides clear boosts in accuracy on various benchmarks.
- The Contrast Vector baseline is interesting and showcases an interesting ability to manipulate the model's honesty as well as other attributes. The demonstration of the lie detector monitoring tool is also interesting, and appears to be simple and practical.

**Weaknesses:**

1. While a lot of details are provided in the appendices, the method section is lacking and doesn't provide enough context to properly understand how the proposed methods are created, or how the method is applied.
2. The authors focus on presenting the results from the honesty study in the main paper, and do a good job at dissecting them. However, providing a summary of the results from the numerous other applications could be helpful.
3. There is no discussion of the limitations of the proposed method.
4. Figure 1 and 2 are not referenced in main text.

**Questions:**

- Given that representation engineering is an emerging approach, what are its current limitations? and what are future direction to improve this approach?

---

> ### Author Response · Authors · 2023-11-21
> **Author Response**
>
> Thank you for your careful analysis of our work. We hope the following response addresses your concerns.
>
> **Clarified method section in main paper.**
>
> We have updated Section 3 in the main paper to give a more detailed and technical description of the methods. Due to the page limit, some details are still left to the appendix, but the main paper is more self-contained now. Thank you for your suggestion.
>
> **Summary of results in appendix.**
>
> We have expanded Section 5 in the main paper to briefly summarize the main results and takeaways from each section in the appendix. Thank you for your suggestion.
>
> **Discussion of limitations.**
>
> We have added a discussion of limitations to Appendix B.1. In particular, our methods require white-box access and thus are not applicable to models only available through APIs. Our methods also require collecting a stimulus set, similar to prior work, although the stimulus set can be automatically generated. Finally, our methods assume that concepts are represented as directions in feature space. This is a common assumption in the interpretability literature, but it may not always hold. Thank you for your suggestion.
>
> **Other points.**
>
> We have added references to Figures 1 and 2 in the main text.

---

### Official Review · Reviewer_RXRb · 2023-11-09

**Soundness:** 2 fair
**Presentation:** 2 fair
**Contribution:** 3 good
**Rating:** 6
**Confidence:** 2

**Summary:**

This work introduces RepE, an approach for interpreting Neural Networks from a top-down perspective through their internal representations. It is two-fold: first, it explores how to read the internal state of a Neural Network, and second, how to control or edit its representation. The approach yields promising results in addressing safety-relevant problems, particularly in the field of Large Language Models (LLMs).

**Strengths:**

* The paper presents a well-motivated approach to enhancing the transparency and reliability of large language models (LLMs). The analogy drawn from neuro-imaging in cognitive science provides a fresh and insightful perspective.
* The proposed method is conceptually straightforward and intuitively appealing.
* The authors thoroughly investigate and improve their method through a series of carefully designed thought experiments and ablation studies. These experiments are both logically sound and firmly grounded in empirical evidence.
* The practical benefits of their method is clear. RepE can clearly improve the transparency of LLMs and prevent them from untruthful responses.

**Weaknesses:**

1. The paper's overall structure and order contribute to its poor readability. The content would benefit from a better balance between technical formality, such as formal definitions or problem formulation, and more intuitive explanations. Currently, the paper leans too heavily on motivations and visualizations.

* Specifically, Figure 2 takes up excessive space with repetitive examples. Moreover, the detailed methodology such as reading and controlling process is relegated to supplementary materials accessible through multiple layers of links. This makes it difficult to access and potentially hinders a reader's ability to gain a comprehensive understanding of the content. I suggest that the authors provide more detailed explanations of key concepts, such as reading vectors, controlling vectors and objective of reading and controlling process, both in a technical manner and in a more general, high-level way.

2. The paper's contributions are not clearly articulated. The Related Works section lacks a comprehensive technical comparison with prior works, making it difficult to distinguish the paper's unique contributions. I recommend that the authors clearly differentiate their work from existing approaches, particularly in terms of their methodology for reading neural networks' internal representations and controlling neural networks' outputs.

3. While the practical benefits of the proposed method are evident, the paper overlooks limitations of the approach. For instance, such as the general disadvantages of a "white-box" approach, such as the need for full accessibility to the LLM's, and the cost of collecting stimulus sets or generating contrastive examples, should be explicitly acknowledged. This limitation is particularly important given that the comparison is conducted against in-context, inference-only LLMs.

**Questions:**

1. Figure 1 (right) is unclear to me, and I did not find a satisfactory explanation in the paper. I would appreciate a clearer explanation from the authors.

2. In Table 1, I wonder the relative effectiveness of LAT compared to prior works on extracting internal representations of neural networks, not to the Heuristic method.

3. Additional related works that also involve research on editing the internal representations or weights of neural networks include [1] and [2]. I hope the authors can explain why these methods were not considered for use in RepE.
* [1] Locating and Editing Factual Associations in GPT.
* [2] Inspecting and Editing Knowledge Representations in Language Models.

4. As mentioned in the Weakness section, I request a more detailed explanation and justification from the authors regarding the potential drawbacks of RepE compared to in-context learning (few-shots) methods. In particular, with reference to Figure 3, I question whether the performance gain over five datasets is significant, especially considering the assumption of accessibility to large language models (LLMs).

**Details Of Ethics Concerns:**

While this work deals with generative models (LLMs), its primary focus is on mitigating the harmful potential of LLMs. Therefore, there are no significant ethical concerns associated with this work.

---

> ### Author Response · Authors · 2023-11-21
> **Author Response (1/2)**
>
> Thank you for your careful analysis of our work. We hope the following response addresses your concerns.
>
> **Improved formal description of method in the main paper.**
>
> While the appendix includes a more extensive and formal definition of our methods, we understand your point and agree that Section 3 in the main paper should use more formal language as well. Thanks to your suggestion, we have updated Section 3 in the main paper to be more straightforward and approachable to readers. We hope you find that this improves readability.
>
> **Importance of Figure 2 for guiding readers.**
>
> As highlighted in the introduction, one of the main contributions in our paper is to demonstrate the potential of representation reading and control methods to address a wide range of safety-relevant problems, which had previously not been explored in related work. Figure 2 gives a high-level overview of all the different tasks we consider, linking to each corresponding section in the caption. Thus, Figure 2 functions as a Table of Contents that gives a preview of the paper. Similar high-level overview figures are common in longer papers and can be quite helpful for readers.
>
> We understand your concern about the balance between discussing technical method contributions and novel application contributions, and we agree that more detailed explanations of key technical concepts are important to include. As discussed above, we have improved Section 3 to address this. We hope this addresses your concern.
>
> **Improved description of technical contributions.**
>
> We have updated the related work in the main paper to focus more on the technical contributions, including the improvements over specific prior methods such as CCS, ITI, and ActAdd. Thank you for your suggestion.
>
> **Discussion of limitations.**
>
> We have added a discussion of limitations to Appendix B.1. As you note, our methods require white-box access and thus are not applicable to models only available through APIs. Our methods also require collecting a stimulus set, similar to prior work, although we point out that the stimulus set can be automatically generated and should be relatively easy to obtain in practice for a given concept. Thank you for your suggestion.
>
> **Comparison between few-shot and LAT in Figure 3 uses the same number of examples.**
>
> We agree that the fairness of comparison between few-shot and LAT in Figure 3 is important. In the caption of Figure 3, we point out that LAT and Few-Shot are compared on equal footing: we use the same set of examples for extracting the direction with LAT and for few-shot examples. In fact, the Few-Shot method has an advantage in that it uses labels whereas our LAT method does not use the labels and is unsupervised. Even so, LAT outperforms Few-Shot.

---

> ### Author Response · Authors · 2023-11-21
> **Author Response (2/2)**
>
> **Reference to the right side of Figure 1.**
>
> Our apologies, this was due to a simple typo. We do actually reference the right side of Figure 1 in the Appendix, but the cleverref package mistakenly labeled it as “Section 1” instead of “Figure 1”. This sentence now reads “On the right side of Figure 1, we visualize…” Thank you for notifying us of this typo.
>
> Specifically, the right side of Figure 1 shows the first two principle components for an intermediate layer of an LLM on a small, custom dataset of high-utility and low-utility sentences. That is, we first create a dataset of sentences describing high-utility and low-utility scenarios, then we pass these sentences through an LLM and record the intermediate activations at a specific layer. We then run PCA on these activations and plot the first two PCs vs the token position.  We also use this dataset to create Figure 10. These are qualitative experiments that we use to illustrate how LLM appear to have representations for the utility concept, which then motivates our quantitative experiments in Appendix D.1.
>
> **Comparison to other methods in Table 1.**
>
> No prior works have explored representation reading for the TruthfulQA MC1 task, so there wasn’t an appropriate comparison for Figure 1. The closest method that could potentially be adapted to this setting is CCS (ITI and ActAdd are representation control methods, not representation reading methods). However, CCS was only evaluated on standard MCQ benchmarks, not TruthfulQA, so we decided to evaluate it in its original settings in Table 8 for a fairer comparison. In Table 1, our goal is to show that representation reading can in fact yield improved accuracy on TruthfulQA, which had not previously been shown. If we have addressed the thrust of your concerns, we kindly ask that you consider raising your score.
>
> **Comparisons ROME and REMIDI.**
>
> The ROME and REMIDI methods are specifically designed for knowledge editing, and we suspect they would outperform our more general-purpose representation control methods. In Appendix E.4, we show qualitative examples of how our representation control methods can be used for knowledge editing. This demonstrates that simply by changing the stimulus set, we can use our method for an entirely different safety-relevant task. However, we do not claim that our methods are state-of-the-art for knowledge editing.

---

> > ### Comment · Reviewer_RXRb · 2023-11-23
> >
> > I appreciate the authors' response addressing my concerns. Although the paper could enhance the clarity of the proposed method's novelty, I find their contribution valuable. Therefore, I have raised my score from 3 to 6.

---

### Meta-Review · Area_Chair_p9hf · 2023-12-11

**Metareview:**

**Summary**: The paper introduces Representation Engineering (RepE), a novel approach for interpreting neural networks, particularly large language models (LLMs), through their internal representations. RepE involves two key aspects: reading the internal state of neural networks and controlling or editing their representations. The approach is grounded in cognitive neuroscience, focusing on population-level representations rather than individual neurons or circuits. The paper presents initial analyses and demonstrates the efficacy of RepE techniques in improving transparency and addressing various safety-relevant problems in LLMs, such as truthfulness and memorization.

**Strengths**: The paper's major strength lies in its innovative approach to enhance the transparency and reliability of LLMs, drawing parallels with neuro-imaging techniques in cognitive science. The method is conceptually simple yet powerful, backed by thorough investigations through thought experiments and ablation studies. It successfully demonstrates practical applications in improving LLM transparency and preventing untruthful responses. The use of representation-focused methods to address safety issues is both novel and significant.

**Weaknesses**: The paper suffers from structural and presentation issues, affecting its readability. It relies heavily on motivations and visualizations, with key methodologies and detailed explanations pushed to supplementary materials. The lack of a clear, technical exposition in the main text hinders comprehension. The paper's contributions could be more distinctly articulated, particularly in distinguishing it from previous works. Additionally, it overlooks discussing the limitations of RepE, such as the need for full access to LLMs and the costs associated with collecting stimulus sets or generating contrastive examples.

**Justification For Why Not Higher Score:**

The paper is interesting and innovative. However, I am uncertain if this type of paper is suitable for ICLR, as the reviewers have pointed out. It may be a better fit for a journal or the newly launched position paper track in ICML 2024. The original version of the paper has several readability issues. Although the authors made some improvements during the rebuttal period, I believe the paper needs to be written accordingly if the authors wish to publish it in a conference.

**Justification For Why Not Lower Score:**

N/A

---

### Decision · Program_Chairs · 2024-01-16

Reject